# Hot topic: Mapping of the human intranasal mucosal thermal sensitivity: A clinical study on thermal threshold and trigeminal receptors

**Susanne Weise** [1]*, **Pauline Hanslik**[1], **Coralie Mignot**[1], **Evgenii Glushkov**[2], **Arnaud Bertsch**[2], **Romain Dubreuil**[3], **Moustafa Bensafi**[4], **Susanne Fuessel**[5], **Thomas Hummel**[1]

1 Smell & Taste Clinic, Department of Otorhinolaryngology, University Hospital Carl Gustav Carus, Technische Universität Dresden, Dresden, Germany, 2 Microsystem Laboratory 4 (LMIS4), School of Engineering, École Polytechnique Fédérale de Lausanne (EPFL), Lausanne, Switzerland, 3 Aryballes Technologies, Grenoble, France, 4 Lyon Neuroscience Research Center, CNRS UMR5292-INSERM U1028-University Claude Bernard Lyon 1, Bron, France, 5 Department of Urology, University Hospital Carl Gustav Carus, Technische Universität Dresden, Dresden, Germany

* Susanne.weise@uniklinikum-dresden.de

## Abstract

### Introduction

The olfactory and trigeminal system are closely interlinked. Existing literature has primarily focused on characterizing trigeminal stimulation through mechanical and chemical stimulation, neglecting thermal stimulation thus far. The present study aimed to characterize the intranasal sensitivity to heat and the expression of trigeminal receptors (transient receptor potential channels, TRP).

### Methods

A total of 20 healthy participants (aged 21–27 years, 11 women) were screened for olfactory function and trigeminal sensitivity using several tests. Under endoscopic control, a thermal stimulator was placed in 7 intranasal locations: anterior septum, lateral vestibulum, interior nose tip, lower turbinate, middle septum, middle turbinate, and olfactory cleft to determine the thermal threshold. Nasal swabs were obtained in 3 different locations (anterior septum, middle turbinate, olfactory cleft) to analyze the expression of trigeminal receptors TRP: TRPV1, TRPV3, TRPA1, TRPM8.

### Results

The thermal threshold differed between locations (p = 0.018), with a trend for a higher threshold at the anterior septum (p = 0.092). There were no differences in quantitative receptor expression (p = 0.46) at the different sites. The highest overall receptor RNA expression was detected for TRPV1 over all sites (p<0.001). The expression of TRPV3 was highest at the anterior septum compared to the middle turbinate or the olfactory cleft. The thermal sensitivity correlated with olfactory sensitivity and results from tests were related to trigeminal function like intensity ratings of ammonium, a questionnaire regarding trigeminal

**Funding:** The Article Processing Charges (APC) were funded by the joint publication funds of the TU Dresden, including Carl Gustav Carus Faculty of Medicine, and the SLUB Dresden as well as the Open Access Publication Funding of the DFG. This project has received funding from the European Union's Horizon 2020 [research and innovation programme] under grant agreement No. 964529 (ROSE = Restoring Odorant Detection and Recognition in Smell Deficits). The funders had no role in study design, data collection and analysis, decision to publish, or preparation of the manuscript.

**Competing interests:** NO authors have competing interests.

function, nasal patency, and $CO_2$ thresholds. However, no correlation was found between receptor expression and psychophysical measures of trigeminal function.

## Discussion

This study provided the first insights about intranasal thermal sensitivity and suggested the presence of topographical differences in thermal thresholds. There was no correlation between thermal sensitivity and trigeminal mRNA receptor expression. However, thermal sensitivity was found to be associated with psychophysical measures of trigeminal and olfactory function.

## Introduction

The olfactory and the intranasal trigeminal systems are closely linked in terms of anatomy and function. Most odors activate the olfactory and trigeminal system simultaneously [1, 2]. Each odor activates the trigeminal system to different degrees, also depending on the odor concentration [2]. This provides the basis for a close relationship between the olfactory and the trigeminal systems.

On the anatomical side, there are links between both systems on different levels such as the nasal mucosa, the olfactory bulb, and specific cerebral areas [3]. In frogs, electrical stimulation of the trigeminal nerve induced a slow potential in the olfactory mucosa [4], emphasizing the interconnectedness of both systems in the periphery. Further, in rats, some trigeminal ganglion cells possess sensory endings in the nasal epithelium and extend branches that directly connect to both the olfactory bulb and the spinal trigeminal complex [5]. Another study in rats suggested that olfactory and trigeminal pathways converge on the same neuronal elements on the central level in the area of the mediodorsal nucleus of the thalamus and that trigeminal input can lead to the modulation of olfactory impressions [6].

Further, the trigeminal and the olfactory systems interact by suppressing and enhancing each other. For example, simultaneous ipsilateral olfactory co-stimulation additionally to trigeminal stimulation increased the chance of localizing a trigeminal stimulus [7]. This relation is also seen the other way around with a dynamic course over time. Anosmic subjects have reduced trigeminal sensitivity [8, 9]. Conversely, a reduced trigeminal sensitivity seems to result in a reduced olfactory function [10]. Furthermore, trigeminal activation is important for the perception of nasal airflow [11] and the detection of possible noxious chemicals [1], which triggers protective respiratory reflexes [12].

Trigeminal nerves can be activated by mechanical, thermal, or chemical stimulation in the head and neck region [13], which has been described to produce the following sensations: sweet, sour, salty, bitter, astringent, cold, fresh, warm, burning, painful, pungent, sharp, scratching, prickling, tickling, "sneeze", or furry [14]. However, it is unclear to which extent taste-related sensations are evoked by chemical stimulations of trigeminal fibers [15].

Anatomically, different branches of the trigeminal nerve innervate different regions in the nose. While the anterior part of the nasal cavity is innervated by the ophthalmic nerve (V1, anterior ethmoidal nerve, infraorbital nerve), the posterior part of the nose is supplied by fibers of the maxillary branch (V2, posterior superior medial nasal nerve, nasopalatine nerve) [16]. The trigeminal system innervates the entire nasal mucosa, while the olfactory system is found mainly in the olfactory cleft, which is in the upper part of the nasal cavity. The respiratory epithelium in the nose is innervated by slower unmyelinated C-fibers (mediating burning, painful

sensations) and faster-myelinated Aδ-fibers (mediating sharp, stinging sensations) [11]. Nociceptors in the nasal mucosa have nerve endings, which are not covered by squamous epithelium. Hence, chemicals can directly activate trigeminal fibers in the mucosa even in the absence of a lesion. Trigeminal chemosensory information is generated mostly due to the activation of receptors belonging to the family of transient receptor potential channels (TRP), which are activated by specific temperature ranges or specific chemicals inducing different types of sensations [11]. The receptor TRPV1 (V1 for vanilloid family 1) is gated, for example, by capsaicin, other vanilloid-containing compounds, acidification, and by an increase in ambient temperature to levels (>43˚C) [17]. An activation of TRPV1 evokes a burning/stinging sensation [18]. The transient-receptor-potential receptor V3 (TRPV3) reacts to temperatures above 39˚C [19] or chemical substances such as thymol [20], which evoke warm sensations. The receptor TRPM8 (M8 for melastatin-8) reacts to temperatures between 8 and 25˚C, or chemicals like menthol or eucalyptol, which evokes a fresh, cooling sensation [18, 21]. In mice, it has been shown that the receptor TRPM8 is involved in the detection of slight temperature changes at the level of body temperature [22]. The receptor TRPA1 (A1 for ankyrin 1) is sensitive to noxious cold below 17˚C [20], which can be perceived as burning pain [23]. Other studies in mice showed that TRPA1 might be involved in the sensation of heat too. Through the gene inactivation of TRPV1/TRPM3/TRPA1 in triple knock-out [KO] mice, it has been shown that the heat sensitivity of the sensory neurons and behavioral responses were abolished, while it was not the case in double KO mice of any combination [24]. The receptor TRPA1 can be activated by different chemicals like bradykinin or menthol [20]. In addition, other acid-sensing ion channel receptors that do not belong to the TRP family are involved in nociception due to acids and sour-tasting substances. Those are voltage-insensitive sodium channels, which could be activated by extracellular protons [25]. Further, there are cholinergic receptors within the trigeminal sensory system which exhibit stereo-selectivity towards R(+)-and S (-)-nicotine [26].

However, the respiratory epithelium is not a homogenous tissue, there are topographic differences regarding the sensitivity to trigeminal stimuli [27–31]. Several studies showed for chemical stimulation the anterior and superior parts of the nasal cavity are more sensitive than the posterior parts [27–29, 31]. Frasnelli et al. showed a higher perceived intensity of $CO_2$ (a selective trigeminal stimulant) at the anterior part of the nose compared to the posterior part (epipharynx). The same pattern was found at the electrophysiological level for EEG-derived event-related potentials (ERPs) which had larger amplitudes for stimuli at the anterior compared to the posterior part [29]. This was further supported by higher response amplitudes at the anterior septum and the lower turbinate compared to the olfactory cleft using an electrophysiological, topical measurement of trigeminally induced activation at the level of the mucosa (negative mucosa potentials, NMP) after presenting $CO_2$ [27]. Similar results were found in 60 subjects by Meusel et al., whereby the highest amplitude for NMP after $CO_2$ application was recorded at the anterior septum. In response to different stimuli like ethanol, menthol, and $CO_2$ differences were detected between the posterior septum and the lateral wall of the posterior nasal cavity [31]. Scheibe et al. (2008) replicated the results in terms of higher amplitudes of negative mucosa potentials at the middle septum compared to the middle turbinate, and the floor of the nasal cavity—including different irritants like $CO_2$, ethyl acetate, and acetic acid. Further, it has been shown that there was an effect of the stimulus concentrations with increased amplitudes of NMP after applying higher concentrations at the middle turbinate and the septum [28]. In contrast, the sensitivity to mechanical stimuli tested with air flow is pronounced in the nasal vestibule [32]. Frasnelli and colleagues (2004) compared EEG-derived event-related potentials and ratings of the sensation in response to gaseous $CO_2$ and mechanical stimuli (air puffs) in 40 subjects in the anterior part of the nose and the posterior

one close to the epipharynx. As mentioned above chemosensory stimuli were perceived stronger in the anterior portion of the nasal cavity, while air puffs did not elicit different sensations of intensity in the anterior or posterior position. On the electrophysiological level, ERP to mechanical stimuli had shorter latencies for stimuli in the anterior nasal cavity [29].

The sensitivity to thermal stimulation has never been studied in the nose. Our study aimed to investigate topographical differences in endonasal thermal sensitivity and trigeminal receptor distribution for temperature sensations based on the quantification of receptor mRNA using quantitative PCR (qPCR).

## Materials and methods

### Participants

A total of 20 healthy volunteers aged 21–27 years (mean 23.1±1.7 years, 11 women) participated in the study from 07/13/2022 to 08/11/2022. Age above 40 years was an exclusion criterion because older age goes along with higher thresholds for trigeminal sensations [32, 33]. Nasal endoscopy by an ENT specialist was done to rule out endonasal diseases such as polyps or any form of chronic rhinosinusitis, which was an exclusion criterion because it can influence trigeminal sensitivity. Further exclusion criteria were: neurological diseases, systemic diseases associated with smell disorders like chronic renal failure, participants with smell impairment, allergic rhinitis, alcohol or drug abuse, and pregnancy. None of the subjects reported acute nasal allergies or other diseases which might affect olfactory function.

The prospective study was conducted at the Smell and Taste Clinic/ Department of Otolaryngology of the Technische Universität (TU) Dresden according to the declaration of Helsinki. It had been approved by the local ethics board (EK 74022022). All participants provided written informed consent and received moderate financial compensation.

### Procedure

In the prior investigation, the participants filled out their medical history including questions on conditions with a possible effect on olfactory function (e.g., nasal surgery, medication, smoking, sinonasal diseases), questionnaires about the subjective importance of olfaction [34], subjective rating of nasal patency and subjective rating of olfactory function.

**Assessment of olfactory and trigeminal function.** Ratings of olfactory function do not strongly correlate with measured olfactory performance [35], hence standardized tests using the "Sniffin' Sticks" (Burghart Messtechnik GmbH, Holm, Germany) were performed, including odor threshold, discrimination, and identification tests [36]. Only participants with normosmia were included in the study as a reduced olfactory function may affect the trigeminal sensation [37].

To test the trigeminal sensitivity patients underwent a lateralization test, rated the intensity of an AmmoLa® stick [38] and were tested for their $CO_2$ threshold [39].

Subjects were screened for trigeminal lateralization [2] using two squeezable bottles which were pressed simultaneously so that they delivered an equal airstream into the participants' nostrils. Only one bottle was filled with an odorant (menthol, 50% v/v diluted in 1,2-propanediol). The visually shielded (sleeping mask) subjects were asked to detect the nostril receiving the odor stimulation using a forced-choice paradigm.

Further, participants were asked to rate the perceived intensity of an AmmoLa® stick, which contains a mixture of ammonium and lavender, which is perceptually dominated by the pungent ammonium. Reduced intensity ratings are accompanied by olfactory dysfunction [38]. In mice, it has been shown that TRPA1 [40] and TRPV1 [41] play a major role in sensing ammonium.

$CO_2$ stimuli were presented with increasing stimulus duration through a bilateral nasal cannula at a low airflow (200 ml/min). In an automated system based on a single staircase paradigm participants received increasing durations of $CO_2$-stimuli (starting at a duration of 100 ms, increased by 100 ms with an interstimulus interval of 10 s [39]). They were asked to press a button when they noticed a stinging or tingling sensation inside the nose. Subsequently, the duration was decreased until there was no perceived sensation and so forth until 7 turning points had been recorded [39]. The threshold was estimated as the mean of the last 4 turning points. All subjects were acoustically shielded (white noise at 65dB SPL via headphones) to prevent hearing any click of the device when $CO_2$ was applied.

Furthermore, all subjects completed a questionnaire regarding their trigeminal function, where they rated their level of agreement with 17 statements such as "Strong pungent smells like fire, vinegar, or acetone evoke strong emotions in me."

**Stimulation.** The Joule heating principle was used to generate the thermal stimuli locally and in a controlled manner. The trigeminal stimulator developed by EPFL consisted of the following parts: a head containing the heating element, a bendable rod (length 19 mm), and a solid holder for electrical connection. The functional part of the stimulator head was fabricated using commercial flexible PCB technology and consisted of a two layers polyimide substrate with gold-coated copper conductive tracks. The heating part was made of a 4 ohm resistance (diameter of stimulation part 3 mm). On the second layer of the flexible PCB, a thermistor was patterned and measured using 4 points method to extract the local temperature during the heating pulses.

The current provided to the heating resistance was kept constant. The temperature was controlled by modulating the time during which the current was flowing in the resistance. The parameters were set by a clinician on a PC through a custom Graphical User Interface and sent to a custom electronic board powered by a 9V battery and consisting of microcontroller board (Arduino Uno) and discrete electronic components. The microcontroller board ensured the duration of the heating pulses as well as the collection of the thermistor measurements and data transmission to the PC and Graphical User Interface.

Nasal endoscopy by an ENT specialist was done to select the side of stimulation as the one without septal deviation. The stimulation was done on the right side in 10 subjects and on the left side in 10 subjects. The participants sat comfortably in a chair keeping the head in a stable position, in an air-conditioned room with temperature kept around 20 to 22°C.

Under endoscopic control, a thermal stimulator was placed in 7 locations: anterior septum, lateral vestibulum, interior nose tip, lower turbinate, middle septum, middle turbinate, and olfactory cleft, see Fig 1. The thermal stimulator was held in place using an external holder, similar to lensless glasses [42]. By positioning the electrode on the intranasal mucosa, discrete mechanical irritation can occur, which might result in a discrete mechanical stimulus. However, this potential irritation was minimized as much as possible through endoscopic positioning by an ENT specialist.

The threshold of the stimulation was tested using a staircase model. Initially, the thermal stimulation was gradually increased until the participant perceived a sensation (first inflection point), then decreased until the perceived sensation disappeared (second inflection point). In this manner, seven inflection points were identified, with the mean calculated from the last 4 inflection points. This procedure aligns with the approach used in the threshold test used for "Sniffin' Sticks" so that 50% of the stimuli could be perceived. Further, stimulation was applied at an intensity above the threshold level (double threshold, duration of 1500ms). Participants were requested to rate each of the perceived stimuli on visual analog scales for pleasantness (0 –"extremely unpleasant" to 5 –"neutral" to 10 –"extremely pleasant"), intensity (0 –"not intense at all" to 10 –"extremely intense"), warmth (0 –"neutral" to 10 –"extremely hot"),

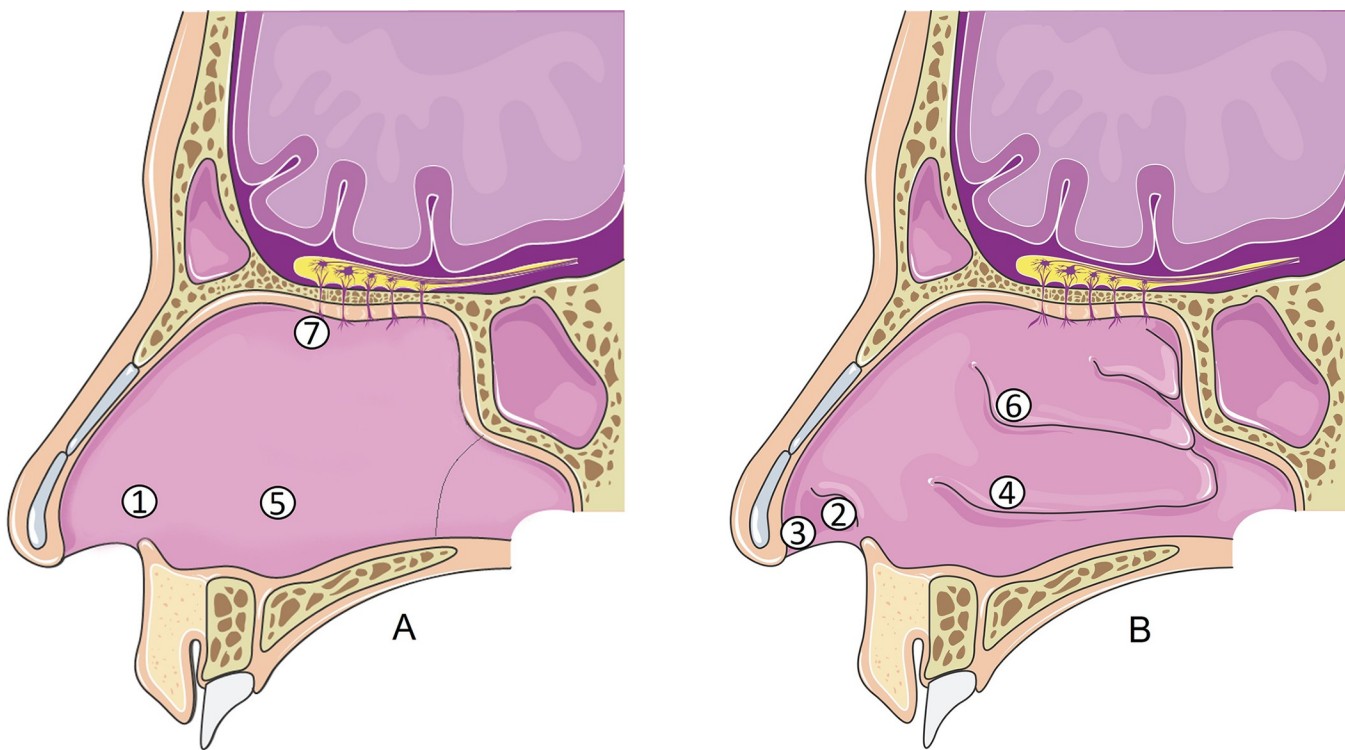

**Fig 1.** Illustration of the stimulation sites showing the longitudinal section of the septum (A) and lateral wall (B) of the nose: anterior septum (1), lateral vestibulum (2), interior nose tip (3), lower turbinate (4), middle septum (5), middle turbinate (6), and olfactory cleft (7). Nasal swabs were taken in anterior septum (1), middle turbinate (6), and olfactory cleft (7). The Figure was modified with markings (letters, numbers) and colour changes after adaptation of "Nasal Cavity" from Servier Medical Art by Servier, licensed under a Creative Commons Attribution 3.0 Unported License".

coldness (0 –"neutral" to 10 –"extremely cold"), tickling sensation (0 –"neutral" to 10 –"extremely tickling"), and pain (0 –"neutral" to 10 –"extremely painful").

**Nasal swabs.**    Under constant endoscopic control, three areas were selected for the nasal brushing on the same side where the thermal stimulation was done using a sterile cotton swab (CLASSIQSwabs™, Brescia, Italy): anterior septum, olfactory cleft, and middle turbinate. There was no nasal decongesting agent or local anesthetics applied to the nasal cavity. The tips of the brushes were stored in RNAlater (Thermo Fisher Scientific, Dreieich, Germany) at -20°C until further processing.

**Analysis of nasal swabs.**    Thawed brush tips were incubated in QIAzol Lysis Reagent (Qiagen, Hilden, Germany) for 30 min at 4°C. After being lysed, RNA from the nasal epithelium samples was isolated using the RNeasy Lipid Tissue Mini Kit (Qiagen). After assessment of RNA quantity and quality by spectrophotometric analysis and automated electrophoresis (2100 Bioanalyzer; Agilent, Waldbronn, Germany), on average, 483 ng of the RNA solution were subjected to cDNA synthesis with SuperScript III Reverse Transcriptase (Thermo Fisher Scientific). This was followed by a preamplification step applying the TaqMan PreAmp Master Mix and gene-specific Taqman Gene Expression assays (all from Thermo Fisher Scientific, see Table 1).

Using the same Taqman assays, mRNA expression levels of the target genes TRPV1, TRPV3, TRPA1, and TRPM8 and two reference genes (Peptidylprolyl isomerase A [PPIA] and TATA-Box binding protein [TBP]) were determined by qPCR on a LightCycler 480 System (Roche Diagnostics, Mannheim, Germany). Manufacturer-validated assays were used, and cultivated according to specified conditions. The qPCR temperature program consisted of the

**Table 1. Taqman assays for qPCR analyses.**

| Gene symbol | Gene name | Assay ID |
|---|---|---|
| PPIA | peptidylprolyl isomerase A | Hs99999904_m1 |
| TBP | TATA-box binding protein | Hs00427620_m1 |
| TRPV1 | transient receptor potential cation channel subfamily V member 1 | Hs00218912_m1 |
| TRPV3 | transient receptor potential cation channel subfamily V member 3 | Hs00376854_m1 |
| TRPA1 | transient receptor potential cation channel subfamily A member 1 | Hs00175798_m1 |
| TRPM8 | transient receptor potential cation channel subfamily M member 8 | Hs01066596_m1 |

following steps: 10 min initial denaturation at 95˚C, 45 cycles of 15 s denaturation at 95˚C, and 1 min annealing/extension at 60˚C. Relative TRPV1, TRPV3, TRPA1, and TRPM8 expression levels to PPIA and TBP were calculated by the 2-ΔΔCT method and used for statistical comparison between the areas.

## Statistical analysis

The data generated and analyzed in the paper can be found in S1 File. Statistical analyses were performed using IBM SPSS Statistics (Statistical Packages for the Social Sciences, version 28.0; SPSS Inc., Chicago, Ill., USA). Descriptive statistics were obtained; continuous variables are expressed as means with standard deviation (SD), while categorical variables are presented as frequencies (percentages). To characterize the perception of the stimulation a Generalized Linear Model (GLM) analyses for repeated measurements were conducted with factors "stimulation strength" (3 steps) and "sites" (7 locations) separately for each descriptor (e.g., intensity, hedonic ratings). GLM analyses were calculated with the target threshold of thermal stimulation, and receptor distribution, respectively.

T-tests for independent samples were performed for the variables of age, odor threshold, discrimination, and identification. Effect sizes are given for significant results as Cohen's d. Chi$^2$ tests were calculated for the effect of absence of TRPM8 and TRPA1 receptors, and sex. The Greenhouse-Geisser adjustment was used to correct in case of violations of sphericity. Pearson correlations were used for the comparison of the thermal threshold and mRNA expression of the receptors. A p-value of $<0.05$ was considered statistically significant.

## Results

### Olfactory and trigeminal function of the studied population

All subjects were characterized as normosmic using Sniffin' Sticks (mean TDI score = 34.0 ±2.2). Eighteen of the 20 participants showed lateralization ability for menthol above chance ($\geq 15$ or $\leq 5$). All participants had a normal range regarding the intensity rating of AmmoLa® (mean = 76.9±17.8, range: 40–100). Further for $CO_2$ the threshold was tested as the following (mean = 1312.4±588.1 ms; range: 199–1966 ms), showing slightly higher values than in a previous study [39].

### Thermal sensitivity

There was a main effect of the factor "site" for the thermal threshold (F(2,99) = 3.63, p = 0.018), see Fig 2. In pairwise comparisons, the anterior septum tended to have a higher thermal threshold compared to the middle turbinate (p = 0.092; $M_{Diff}$ = 490.3, 95%-CI[41.2, 1021.7]). Even if the two subjects who could not lateralize menthol odor were excluded, the

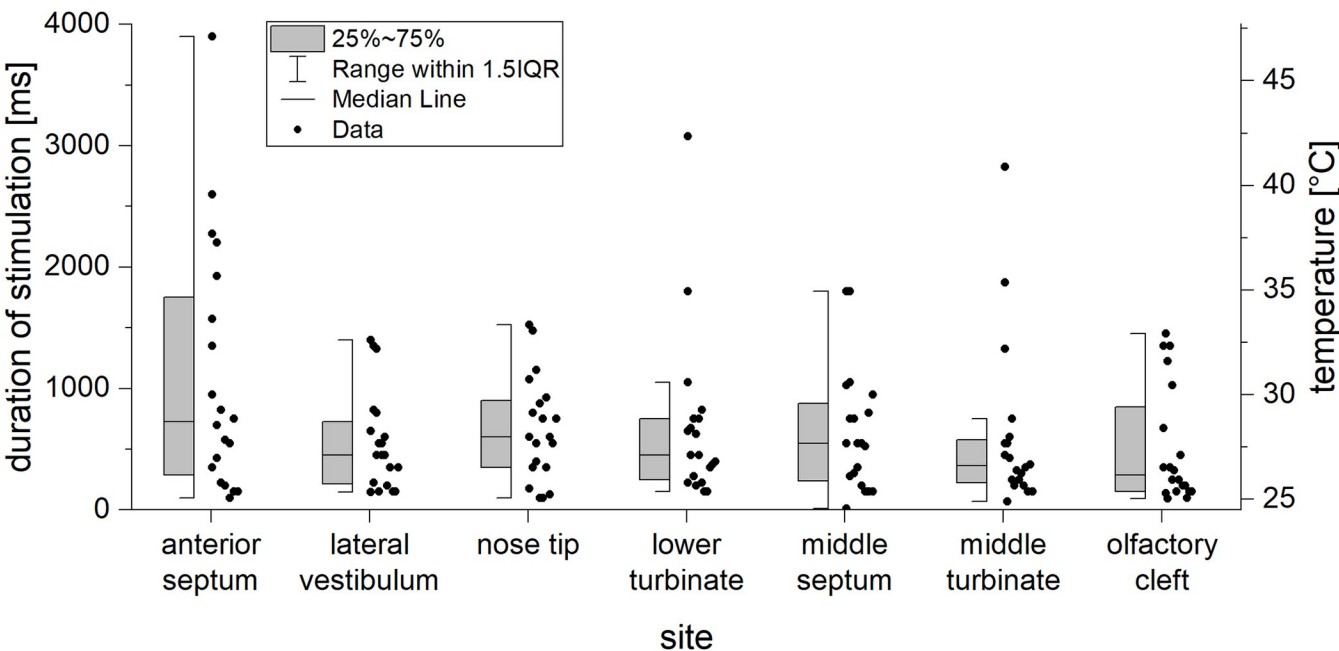

**Fig 2. Thermal threshold in different sites of the nose regarding the temperature at the right site and the duration of stimulation at the left site: The anterior septum tended to have a higher thermal threshold.**

slight but non-significant difference in the thermal threshold between the different locations persisted (F(2,83) = 2.77, p = 0.055).

## m-RNA expression of trigeminal receptors

There was no main effect for the factor "site" on the mRNA expression of receptors (F(2) = 0.80, p = 0.46), but a main effect for the factor "receptor type" (F(1,13) = 376.81, p<0.001) with no interaction between factors "site" and "receptor type" (F(1,94) = 0.86, p = 0.43). This suggested that all receptors were found with similar density at the different sites, with a significantly higher expression of receptor TRPV1 followed by TRPV3, TRPA1, and TRPM8 (Fig 3).

The relative mRNA expression levels of the receptors were similar for both reference genes, so only the results normalized to TBP are reported in this section. For the results normalized to PPIA, please see S1 File.

For the relative expression levels of TRPV1, there was no effect for the site (F(2) = 0.052; p = 0.95) and no difference in the pairwise comparison (p = 1.00 for all pairs).

In contrast, there was a difference in the relative expression of TRPV3 regarding the site (F(1.03) = 10.40; p = 0.004), with a higher expression of TRPV3 at the anterior septum compared to the middle turbinate (p = 0.011; $M_{Diff}$ = 0.036, 95%-CI[0.007–0.064]) or olfactory cleft (p = 0.016, $M_{Diff}$ = 0.033, 95%-CI[0.006–0.061]), see Fig 3.

The receptor TRPA1 was often not detectable in all 3 sites (anterior septum 85%, middle turbinate 40%, and olfactory cleft 60%), with a difference in the distribution ($\chi^2$(2) = 8.60, p = 0.014 For the relative TRPA1 expression there was no effect for the site (p = 0.88), nor a difference in the pairwise comparison (p = 1.00 for all pairs).

Regardless of the receptor site, TRPM8 was often not detectable (anterior septum 70%, middle turbinate 70%, and olfactory cleft 60%, $\chi^2$(2) = 0.60, p = 0.74). Regarding the relative TRPM8 expression, there was no effect for the site (p = 0.85).

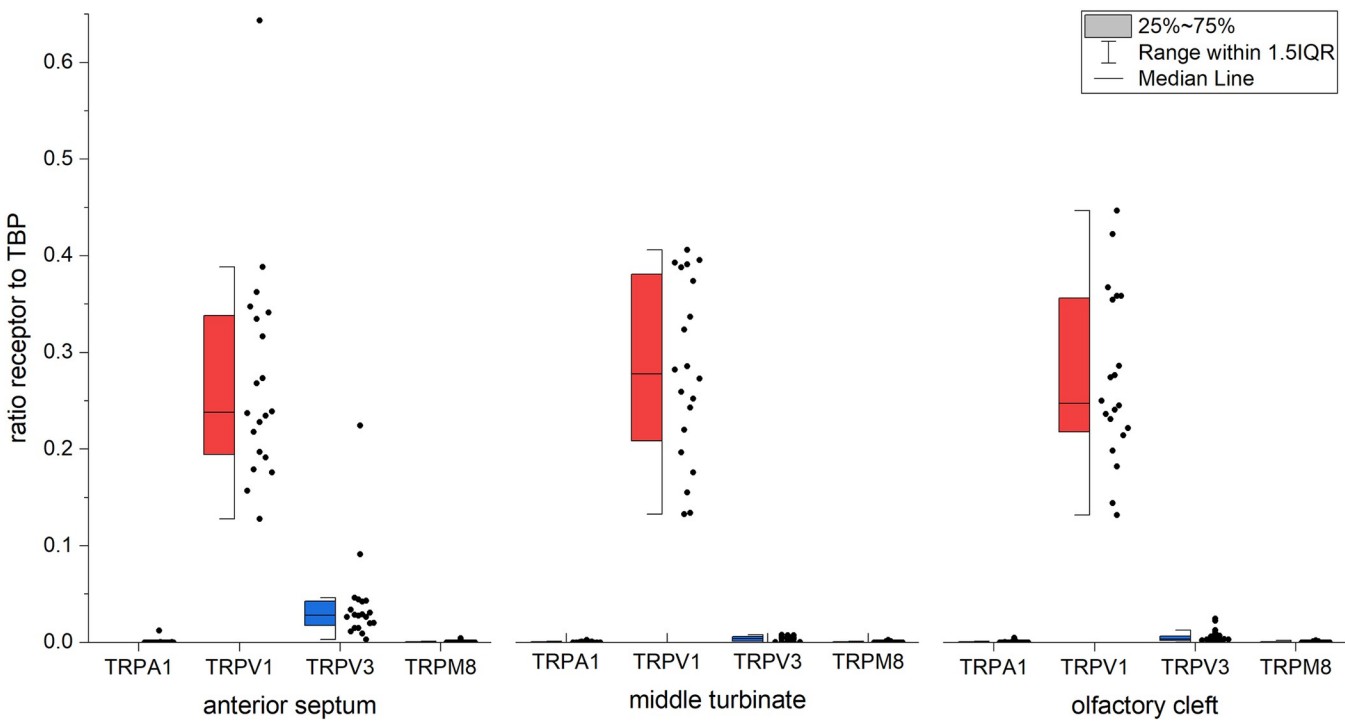

**Fig 3. Relative receptor mRNA expression normalized to TBP over the sites.**

## Correlation of thermal sensitivity and receptor expression levels with psychophysical tests

For **thermal thresholds,** there was a positive correlation between several sites (see Fig 4), as the thermal threshold at the anterior septum and the nose tip (r = 0.46, p = 0.040), lower turbinate (r = 0.45, p = 0.045), middle septum (r = 0.61, p = 0.004), middle turbinate (r = 0.76, p<0.001) and olfactory cleft (r = 0.64, p = 0.002). Moreover, there was a positive correlation between the thermal threshold at the lateral vestibulum and the nose tip (r = 0.49, p = 0.027), respectively the lower turbinate (r = 0.53, p = 0.016). Further, a positive correlation emerged between the thermal threshold at the nose tip and the lower turbinate (r = 0.58, p = 0.008), middle septum (r = 0.51, p = 0.020), middle turbinate (r = 0.58, p = 0.008), and the olfactory cleft (r = 0.50, p = 0.024). Also, a positive correlation between the thermal threshold at the lower turbinate and the middle septum (r = 0.67, p = 0.001) and the middle turbinate (r = 0.72, p<0.001) was found. In addition, the thermal threshold at the middle septum and the middle turbinate (r = 0.85, p<0.001) respectively the olfactory cleft (r = 0.74, p<0.001) showed a positive correlation. Further, the thermal threshold for the middle turbinate and the olfactory cleft (r = 0.69, p<0.001) had a positive correlation.

Within the **relative receptor expression levels,** there was a positive correlation between TRPV3 at the anterior septum and TRPA1 (r = 0.77, p<0.001) at the olfactory cleft respectively TRPM8 (r = 0.46, p = 0.042) at the olfactory cleft, see Fig 4. Further, there was a negative correlation at the middle turbinate for TRPV3 and TRPV1 (r = -0.61, p = 0.004). For the olfactory cleft, a positive correlation was shown for TRPM8 and TRPA1 (r = 0.54, p = 0.014).

There was a positive correlation found for the **relative TRPV3 expression** at the olfactory cleft and the **thermal threshold** at the olfactory cleft (r = 0.60, p = 0.005), middle turbinate (r = 0.80, p<0.001), middle septum (r = 0.59, p = 0.006), lower turbinate (r = 0.47, p = 0.037),

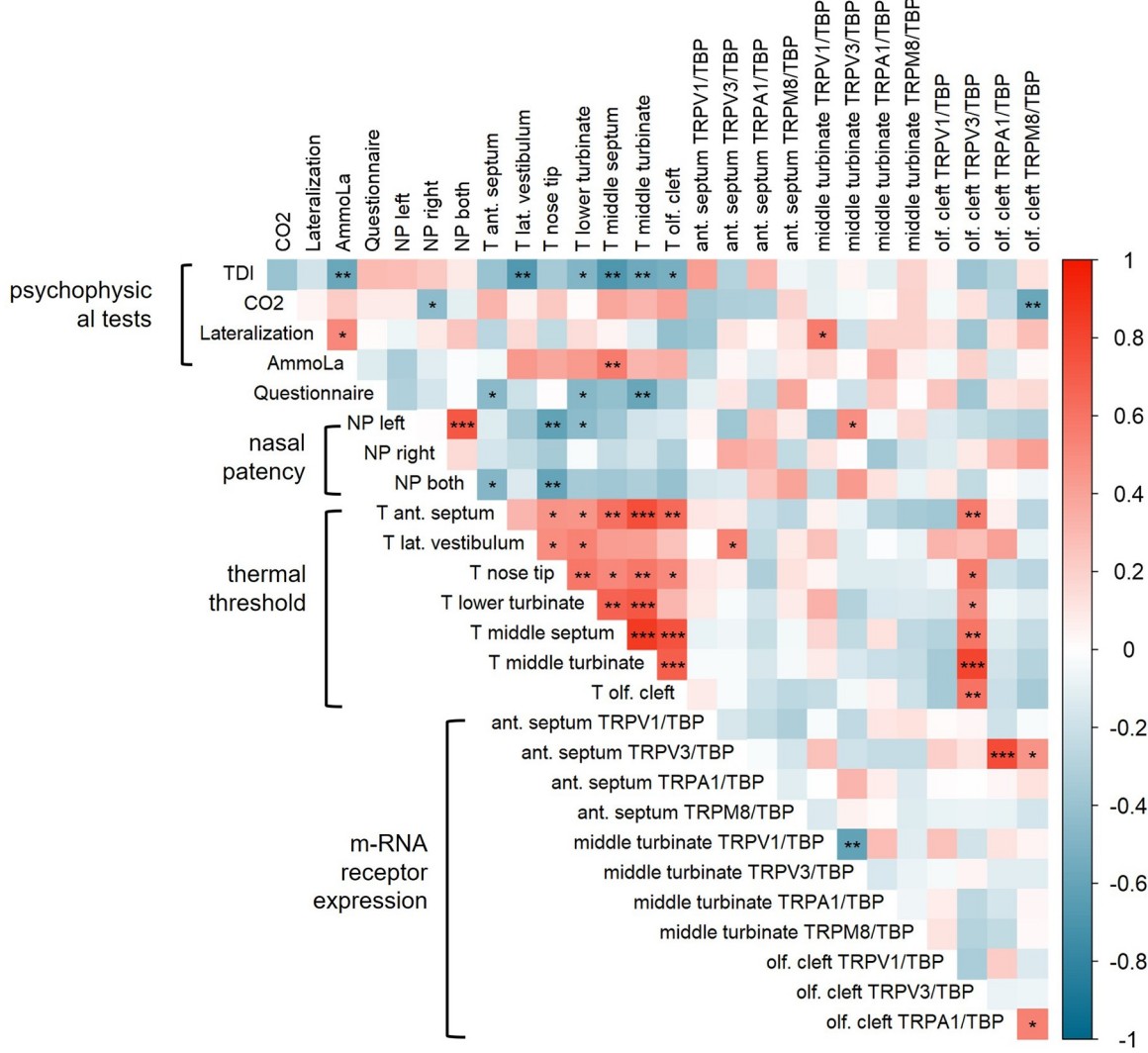

**Fig 4. Pearson's correlation represented as a multidimensional heatmap.** Pearson's correlation plot visualized the correlation values between 26 acquired parameters (psychophysical tests, trigeminal questionnaire, nasal patency, thermal threshold, and m-RNA receptor expression). Scale bar represents the range of the correlation coefficients displayed. Red color (1.00–0.70) indicated a strong positive correlation, blue color (−0.70–−1.00) revealed a strong negative correlation, light color above 0.40 or under −0.40 indicated moderate correlation, grey color with a coefficient of approximately 0 displayed no correlation. Significant p-values are shown with an asterisk*, the classification is as follows: p≤0.05 *, p≤0.01 **, p≤0.001 ***.

nose tip (r = 0.55, p = 0.012), and anterior septum (r = 0.57, p = 0.009), see Fig 4. Paradoxically, this means that a higher thermal threshold, i.e., a lower thermal sensitivity, was accompanied by a higher receptor expression. Further, there was a positive correlation between the thermal threshold at the lateral vestibulum and the relative expression of TRPV3 at the anterior septum. For other sites, there was no correlation between the receptor mRNA expression and the thermal threshold.

The **TDI score** exhibited a negative correlation with the thermal threshold at the lateral vestibulum (r = -0.672, p = 0.001), the lower turbinate (r = -0.492, p = 0.028), the middle septum (r = -0.678, p = 0.001), middle turbinate (r = -0.57), the olfactory cleft (r = -0.53, p = 0.016) and the mean value for thermal threshold (r = -0.642, p = 0.002), indicating that a higher TDI score

was associated with a lower thermal threshold respectively a higher thermal sensitivity (see Fig 4). There were no correlations found between the receptor levels and TDI, but a trend of a negative correlation between TDI and the m-RNA expression of TRPV3 at the olfactory cleft (p = 0.086).

Further, a trend of a positive correlation between the **$CO_2$ threshold** and the thermal threshold at the olfactory cleft (r = 0.404, p = 0.077) and at the middle septum (r = 0.382, p = 0.096) was shown, see Fig 4. Except for the negative correlation of receptor expression for TRPM8 at the olfactory cleft and the $CO_2$ threshold (r = -0.580, p = 0.007), there were no correlations found for the receptor levels.

There was a trend of a negative correlation between the **lateralization test** and the thermal threshold at the olfactory cleft (r = -0.420, p = 0.065). Regarding the lateralization and the receptor expression a positive correlation for TRPV1 at the middle turbinate (r = 0.560, p = 0.010), a trend for the TRPV1 at the anterior septum (r = -0.381, p = 0.097), and a trend of a negative correlation for TRPV3 at the olfactory cleft (r = 0.385, p = 0.094) appeared.

A trend of a positive correlation was found between the intensity ratings of the **AmmoLa®** stick and the thermal threshold at the lateral vestibulum (r = 0.440, p = 0.052), the nose tip (r = 0.381, p = 0.097), as well as the lower turbinate (r = 0.429, p = 0.059). A positive correlation emerged between the intensity ratings of the AmmoLa® stick and the thermal threshold at the middle septum (r = 0.565, p = 0.009). However, there was no correlation between the mean value of thermal threshold (for all sites) and the intensity ratings of the AmmoLa® stick. There was no correlation between the receptor m-RNA expression and AmmoLa®.

Further, a positive correlation between the **nasal patency at the left side** and the relative receptor expression of TRPV3 at the middle turbinate (r = 0.48, p = 0.031) was shown.

For the **trigeminal questionnaire,** a negative correlation emerged to the thermal threshold at the anterior septum (r = -0.46, p = 0.040), lower turbinate (r = -0.47, p = 0.038), and middle turbinate (r = -0.59, p = 0.006).

## Evaluation of sensation

The participants assessed the thermal stimulation across different types of stimulation (threshold, doubled threshold, and stimulation duration of 1500 ms) and different locations of stimulations as comparable using visual analog scales. The perceived sensation was described as slightly painful, slightly ticklish, slightly warm, not cold, less intense, and somewhat unpleasant across all stimulation areas, as presented in Table 2.

Comparing the ratings regardless of the stimulation site, a main effect was found for the stimulation strength (p = 0.017, F(2) = 4.59) and an interaction effect between the stimulation strength and the rating of the descriptors (p = 0.04, F(4.6) = 2.52). In a pairwise comparison, the stimulation at the level of threshold was perceived as more painful (p = 0.013, $M_{Diff}$ = 0.43, 95%-CI[0.08, 0.77]) and more intense (p = 0.001, $M_{Diff}$ = 0.56, 95%-CI[0.23, 0.89]) than at the doubled threshold.

Analyzing the stimulation sites separately the following results emerged: a main effect of the stimulation site was found (p<0.001, F(6) = 6.08), showing a difference in the pairwise

**Table 2. Evaluation of the sensation regarding intensity, pain, tickling warmth, cold, and pleasantness at different stimulations.**

| Stimulation | Stimulation duration (ms) | Intensity (VAS 0–10) | Pain (VAS 0–10) | Tickling (VAS 0–10) | Warmth (VAS 0–10) | Cold (VAS 0–10) | Pleasantness (VAS 0–10) |
|---|---|---|---|---|---|---|---|
| Threshold | 674.7±477.7 | 3.7±1.1 | 2.4±0.9 | 2.4±1.3 | 1.7±1.4 | 0.6±0.6 | 3.4±1.1 |
| Doubled threshold | 1285.6±782.1 | 3.1±1.2 | 2.0±1.1 | 2.1±1.5 | 1.3±1.1 | 0.7±0.7 | 3.5±1.1 |
| 1500ms | 1500 | 3.4±1.3 | 2.1±1.3 | 2.2±1.5 | 1.6±1.3 | 0.6±0.7 | 3.6±1.1 |

**Table 3. Pairwise comparison of significant differences in the sensation regarding the stimulation site.**

| Sensation | Stimulation site | p-value | MDiff | CI |
|---|---|---|---|---|
| Pain | olfactory cleft—anterior septum | 0.007 | 1.72 | 0.34, 3.10 |
| | olfactory cleft—lateral vestibulum | 0.032 | 1.75 | 0.92, 3.42 |
| | olfactory cleft—nose tip | 0.042 | 1.97 | 0.40, 3.89 |
| | olfactory cleft—lower turbinate | 0.024 | 2.00 | 0.17, 3.83 |
| Intensity | anterior septum—middle septum | 0.016 | 1.14 | 0.15, 2.14 |
| | lateral vestibulum—olfactory cleft | 0.033 | 1.42 | 0.07, 2.77 |
| Hedonic | anterior septum—olfactory cleft | 0.013 | 1.46 | 0.21, 2.70 |
| | lateral vestibulum—middle septum | 0.005 | 1.14 | 0.26, 2.02 |
| | lateral vestibulum–middle turbinate | 0.042 | 1.21 | 0.03, 2.40 |
| | lateral vestibulum—olfactory cleft | 0.002 | 1.77 | 0.55, 3.00 |
| | nose tip—middle septum | 0.017 | 1.14 | 0.14, 2.14 |
| | nose tip—middle turbinate | 0.037 | 1.21 | 0.04, 2.38 |
| | nose tip—olfactory cleft | <0.001 | 1.77 | 0.81, 2.74 |
| | lower turbinate—olfactory cleft | 0.037 | 1.58 | 0.06, 3.10 |

comparison between the posterior structure of olfactory cleft, and the more anterior structures like the anterior septum, lateral vestibulum, nose tip, and lower turbinate, see Table 3. For the perceived intensity there was a significant effect of the site of stimulation (p = 0.001, F(3,86) = 5.15) with a significant difference in the pairwise comparison between the anterior septum and the middle septum and between the lateral vestibulum and the olfactory cleft, indicating a more intense perception at the posterior parts, see Table 3. The perception of hedonic showed a main effect for the site of stimulation (p<0.001, F(3.72) = 10.0). In the pairwise comparison we found a more unpleasant sensation in the posterior part compared to the anterior ones, see Table 3. For the sensation of tickling, there was a main effect for the stimulation site (p = 0.036, F(3.2) = 2.98). In the pairwise comparison, there were no significant differences. For the perception of coldness, there were no main effects for the stimulation site (p = 0.50, F(3.59) = 0.83). For the sensation of warmth, no main effects were found for the stimulation site (p = 0.51, F(6) = 0.89.

## Discussion

In this functional and behavioral study including young, healthy patients without an olfactory dysfunction, the following main results emerged. **First**, using a thermal heating device the thermal sensitivity can be measured intranasally. **Second**, the thermal threshold differed between sites (p = 0.018), with a trend of a higher threshold at the anterior septum (p = 0.092). **Third**, at different sites there were no differences in the trigeminal receptor expression levels (p = 0.46). However, the receptor types were expressed differently (p<0.001), showing the highest overall m-RNA expression in the nasal mucosa for TRPV1 over all sites, followed by TRPV3, TRPA1, and TRPM8. **Fourth**, analyzing the receptors separately, the expression of TRPV3 at the anterior septum was higher compared to the middle turbinate or olfactory cleft. **Fifth**, for the thermal sensitivity a correlation was shown to TDI-score and trigeminal tests like AmmoLa® stick, a questionnaire regarding trigeminal function, and nasal patency. Sixth, a correlation was found between thermal sensitivity and receptor expression of TRPV3. Seventh, the sensation of the stimulation was perceived as more intense, more painful, and less pleasant in the posterior parts compared to the anterior ones.

In this clinical study including 20 healthy subjects, a thermal heating device was used for the first time intranasally to measure the local thermal sensitivity. Existing literature has primarily focused on characterizing trigeminal stimulation through mechanical and chemical stimulation parts [27–32, 43, 44], neglecting thermal stimulation thus far. This study sheds light on this area by providing a characterization at both functional and physiological aspects.

According to the present data, the thermal sensitivity in the nose differed between different sites. Our findings on thermal sensitivity align with prior research on trigeminal sensitivity in the nose. Previous studies have indicated variations in intensity ratings [29], recordings from the epithelium [31], and cortical event-related potentials [29] based on the sites of stimulation. Thermal stimulation showed a tendency for higher thresholds in the anterior septum. Interestingly, the chemosensory sensitivity, in contrast, was highest at the (anterior) septum, measured using NMPs for $CO_2$, ethyl acetate, and acetic acid [27, 28, 31]. Additionally, mechanical stimulation revealed a more sensitive perception in anterior parts of the nose like the nasal vestibule than the rest of the nasal mucosa [32]. However, Frasnelli et al. (2004) could not find different intensity ratings regarding the stimulation site (behind nasal valve vs. epipharynx) for air puffs as mechanical stimulation although there were differences in chemosomatosensory event-related potentials [29]. Thus, the local perception of thermal stimuli could be different from other trigeminal stimuli. The intranasal trigeminal stimuli appear to elicit different activations depending on the nature of stimulus and the site of stimulation. The trigeminal system is supposed to keep potentially dangerous substances out of the body, and the nature of the stimulus could have a significant influence on this. In further studies, thermal measurements should be done especially at more posterior areas of the nasal cavity to evaluate the sensitivity in the whole nasal cavity. It is conceivable that due to the higher body temperature in the internal body, such as in the epipharynx, higher temperatures would need to be applied to notice a difference in more posterior regions. This could correspond to the posterior stimulation being perceived as more intense, painful and unpleasant. Furthermore, the thermal stimuli were applied by an electrode, so that a discrete mechanical irritation could occur, which might result in a simultaneous discrete mechanical stimulus. This could have biased the perception of the stimuli. In future studies, it would be intriguing to explore thermal stimulation without concurrent mechanical stimulation, such as using a laser, and utilizing a larger stimulating electrode to target larger areas of the intranasal mucosa.

Most studies on thermal sensation refer to rodents [45]. The topographical distribution of trigeminal receptors in the human nasal cavity has been mainly studied indirectly due to different substances which stimulated specific receptors without testing the receptor itself [27–32]. Consistent with Poletti et al. [46], the highest receptor density was found for TRPV1 and the lowest receptor expression for TRPA1 and TRPM8 showing a high rate of missing receptors on all sites. In contrast to Poletti et al., who showed higher TRPV1 expression in posterior areas, we could not demonstrate such a difference and found an almost uniform expression of TRPV1. However, Poletti et al. collected samples more posteriorly (anterior sphenoid sinus wall, posterior septum, etc.) potentially allowing for the detection of differences. In contrast, our study mainly focused on samples from the anterior half of the nasal cavity, thus making a comparison of the TRPV1 expression between anterior and posterior areas remains unclear. At the functional level, previous studies demonstrated that chemical stimulation with $CO_2$ elicited higher NMP (negative mucosa potentials) amplitudes [27, 31] and more pronounced ERPs at the level of central processing [29] with anterior stimulation compared to a more posterior one. The second most commonly expressed receptor was TRPV3, which was expressed more frequently in the anterior septum compared to the middle turbinate or olfactory cleft. The receptor TRPV3 exhibits a distinct threshold, activating at innocuous (warm) temperatures and responding more strongly to noxious temperatures. The difference in receptor

expression according to the site may arise from the protective role of trigeminal sensitivity, an increased sensitivity at the entrance of the nose could have a protective effect. The TRPV1 receptor is specifically expressed in keratinocytes, indicating that skin cells possess the ability to detect heat through molecules akin to those found in heat-sensing neurons [19]. However, the different receptors involved in the sensation of heat suggest redundancy to heat processing by other receptors [24, 45]. Lee et al. showed in rats that besides TRPs, other receptors like calcitonin gene-related peptide (CGRP) were found on the septal mucosa, the ventromedial side of the nasoturbinates, and the dorsal surface of the maxilloturbinates, but more rarely on the lateral side of the naso- and maxilloturbinates and on the lateral nasal wall. Further, the intraepithelial CGRP fibers were denser in anterior areas compared to the posterior ones [47]. In addition, studies in mice and rats demonstrated widespread immunoreactive epithelial cells in the medial septum and lateral walls of the anterior nasal cavity as well as chemosensory cells reaching the surface of the nasal epithelium forming synaptic contacts with trigeminal nerve fibers. The chemosensory transduction in the chemosensory is done by T2R "bitter-taste" receptors among others. On the functional level, it has been shown that bitter substances can influence respiratory flow through trigeminal activation [48]. The question of which receptors play a critical role in temperature perception in humans, apart from thermo-TRPs, has not been definitively resolved.

Additionally, there might be an additional influence of the central nervous system in sensing temperature [45]. In mice thermo-TRPs, e.g. TRPM8, have been found in central structures like the hypothalamus and other limbic structures [49]. Further studies should address the cortical processing of thermal stimuli.

We acknowledge that the method of qPCR to examine the trigeminal receptors has some limitations. This method provided information about the mRNA expression of the receptors concerning two reference genes examined, but not regarding the active proteins that might influence any function. Therefore, in further studies the protein function of trigeminal receptors needs to be addressed. Also, sample collection through swabs primarily reaches the superficial layers, so that receptors that may lie in the depth of the tissue cannot be detected. In a previous study on biopsies of the oral mucosa, the receptors TRPA1 and TRPM8 have been found in the deeper layer of lamina propria [50]. This structure might be more difficult to reach through the swabs, which could explain the higher absence ratio of those receptors.

This study disregards the influence of age. Only young, healthy subjects without any olfactory dysfunction were included. Older patients are known to have an increased oral thermal detection threshold [51] due to reduced thermoreceptor density and superficial skin blood flow. Animal studies also provide evidence for a possible change in the peripheral nervous system with a decreased conduction velocity, and fiber loss due to aging [52]. Certain diseases also seem to be associated with the altered expression of the TRPV3 receptors, for example [53].

Higher TDI score were associated with lower thermal thresholds respectively higher thermal sensitivity. This might emphasize the close interlink of the olfactory and trigeminal system, which has been shown anatomically on different levels such as the nasal mucosa, the olfactory bulb, and specific cerebral areas [3, 9]. In rats, an interconnection of trigeminal ganglion cells in the nasal epithelium has been shown with branches reaching directly the spinal trigeminal complex and the olfactory bulb. [5] The thermal threshold, as a component of the trigeminal function, exhibited a close relationship with other trigeminal tests like AmmoLa® stick, a questionnaire regarding trigeminal function, and nasal patency. The thermal sensitivity could thus be an important aspect for characterizing the trigeminal function. The questionnaire utilized in this study could serve as a valuable tool for evaluating trigeminal function. A follow-up study is currently being planned to establish its efficacy further. Within the thermal sensitivity at different areas, a positive correlation was found. The more sensitive the thermal

threshold was at one point, the more sensitive it was at other examination sites as well. In a pairwise comparison only a trend of a slightly higher threshold at the anterior septum was identified indicating a similar thermal threshold at different sites. However, this study did not include more posterior areas of the nose. In the present study, thermal sensitivity was investigated intranasally in humans for the first time, so there are no comparative studies to date.

It would be highly interesting to analyze the sensitivity to cold temperatures intranasally. Dedicated thermoreceptors consist of unmyelinated C-fiber axons, while cooling-responsive afferents with thinly myelinated Aδ-axons have also been observed [54]. The physiology of thermal perception could result in a higher sensitivity to cold stimuli in the nose, which should be investigated in further studies. This could also be associated with the function of warming the breathing air.

The receptor expression and the thermal sensitivity were analyzed by correlation and characterized using olfactory and trigeminal function. There was a positive correlation between the expression of TRPV3 at the anterior septum and the thermal threshold at the lateral vestibulum. Further, a positive correlation was shown between the TRPV3 at the olfactory cleft and the thermal threshold at all sites (except the lateral vestibulum). However, the expression of the TRPV3 receptor is responsible for heat processing. A higher expression of the receptor should be accompanied by a lower thermal threshold, which would be contrary to our findings. Nevertheless, the used method might be not adequate concerning differences in active proteins.

In summary, the expression of the TRP receptors as well as the thermal sensitivity at the different sites would have to be investigated in a larger sample size (possibly in a multicentric design) including further methods referring to protein function or histological sections, to analyse the relationship between structure and function.

The present results highlight for thermal trigeminal stimulation that the respiratory mucosa is not a homogeneous tissue in terms of its sensitivity to irritants.

## Conclusion

Our ability to sense temperature is crucial for avoiding potential hazards, especially in the upper airways. This clinical study was the first to investigate intranasal thermal sensitivity in humans and related it to other trigeminal and olfactory functions. The thermal threshold differed between sites intranasally with a trend of a higher threshold at the anterior septum. The thermosensitive receptors were expressed to a variable extent with the highest expression for TRPV1 followed by TRPV3, TRPA1, and TRPM8. However, they were mainly similarly distributed in the nose, except for the higher expression of TRPV3 at the anterior septum. There was a correlation between thermal sensitivity and olfactory function, ratings of trigeminal stimuli (AmmoLa® stick), a questionnaire regarding trigeminal function, nasal patency, and $CO_2$ threshold.

Future studies should address the function of the proteins of thermosensitive-TRPs, differences regarding the thermal sensitivity in patients with a reduced trigeminal function analyzing a larger sample size (in a multicentric design), including cold stimulation.

## Supporting information

**S1 File.**
(DOCX)

## Acknowledgments

We are indebted to Andrea Lohse-Fischer for performing the qPCR measurements and Nicole Power Guerra for her help with parts of the statistical analyses.

## Author Contributions

**Conceptualization:** Susanne Weise, Moustafa Bensafi, Thomas Hummel.

**Data curation:** Susanne Weise, Pauline Hanslik, Coralie Mignot.

**Formal analysis:** Susanne Weise, Coralie Mignot, Susanne Fuessel, Thomas Hummel.

**Funding acquisition:** Moustafa Bensafi.

**Investigation:** Pauline Hanslik, Evgenii Glushkov, Arnaud Bertsch, Romain Dubreuil.

**Methodology:** Susanne Weise, Evgenii Glushkov, Arnaud Bertsch, Romain Dubreuil, Moustafa Bensafi, Susanne Fuessel, Thomas Hummel.

**Project administration:** Thomas Hummel.

**Supervision:** Thomas Hummel.

**Writing – original draft:** Susanne Weise.

**Writing – review & editing:** Susanne Weise, Pauline Hanslik, Coralie Mignot, Evgenii Glushkov, Arnaud Bertsch, Romain Dubreuil, Moustafa Bensafi, Susanne Fuessel, Thomas Hummel.

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
