## [Editor Report · Decision Letter 0]

13 Nov 2023

PONE-D-23-24390Mapping of the human intranasal mucosal thermal sensitivity: A clinical study on thermal threshold and trigeminal receptorsPLOS ONE

Dear Dr. Weise,

Thank you for submitting your manuscript to PLOS ONE. Please revise your current manuscript by including the figure legends and upload a new version with the figures also. Please double check that, once uploaded, the manuscript is complete.

We look forward to receiving your revised manuscript.

Kind regards,

Johannes Reisert

Academic Editor

PLOS ONE

Journal Requirements:

"The Article Processing Charges (APC) were funded by the joint publication funds of the TU Dresden, including Carl Gustav Carus Faculty of Medicine, and the SLUB Dresden as well as the Open Access Publication Funding of the DFG.

This project has received funding from the European Union's Horizon 2020 [research and innovation programme] under grant agreement No. 964529 (ROSE = Restoring Odorant Detection and Recognition in Smell Deficits)."

6. Please include a separate caption for each figure in your manuscript.

7. Please ensure that you refer to Figure 4 in your text as, if accepted, production will need this reference to link the reader to the figure.

8. We note you have included a table to which you do not refer in the text of your manuscript. Please ensure that you refer to Table 5 in your text; if accepted, production will need this reference to link the reader to the Table.

---

## [Author Response · Author response to Decision Letter 0]

7 Dec 2023

Dear Reviewer,

I appreciate the time and effort you and the reviewers have dedicated to evaluating my manuscript. I sincerely apologize for the formatting errors present in the submitted manuscript titled "Mapping of the human intranasal mucosal thermal sensitivity: A clinical stud on thermal threshold and trigeminal receptors". Unfortunately, there were unintended issues that led to error messages and the absence of figure captions.I have carefully considered the points raised and would like to provide a detailed response to each.

1. Change of Figure Legends: The reviewers suggested a change in the figure legends. I have revisited the legends and made the necessary adjustments to ensure clarity and accuracy. 

2. Upload of a New Version with Figures: I have carefully reviewed all the figures in the manuscript and have made the required modifications. 

I would like to express my gratitude for the constructive feedback provided by the reviewers. Their insights have been invaluable in enhancing the quality of the manuscript.

Thank you for your time and consideration. I look forward to hearing your thoughts on the revised manuscript.

Best regards,

Susanne Weise

---

## [Decision Letter · Decision Letter 1]

9 Jan 2024

PONE-D-23-24390R1Hot topic: Mapping of the human intranasal mucosal thermal sensitivity: A clinical study on thermal threshold and trigeminal receptorsPLOS ONE

Dear Dr. Weise,

Thank you for submitting your manuscript to PLOS ONE. After careful consideration, we feel that it has merit but does not fully meet PLOS ONE’s publication criteria as it currently stands. Therefore, we invite you to submit a revised version of the manuscript that addresses the points raised during the review process. Please submit your revised manuscript by Feb 23 2024 11:59PM. If you will need more time than this to complete your revisions, please reply to this message or contact the journal office at plosone@plos.org. Please include the following items when submitting your revised manuscript:A rebuttal letter that responds to each point raised by the academic editor and reviewer(s). You should upload this letter as a separate file labeled 'Response to Reviewers'.A marked-up copy of your manuscript that highlights changes made to the original version. You should upload this as a separate file labeled 'Revised Manuscript with Track Changes'.An unmarked version of your revised paper without tracked changes. You should upload this as a separate file labeled 'Manuscript'.

We look forward to receiving your revised manuscript.

Kind regards,

Johannes Reisert

Academic Editor

PLOS ONE

Additional Editor Comments:

Please address all the comments made by the reviewer by elaborating in particular on your experimental procedures in regards to thermal stimulation, clarify mechanical vs thermal stimulation and your positive controls for PCR experiments

Reviewers' comments:

Reviewer's Responses to Questions

**Comments to the Author**

1. If the authors have adequately addressed your comments raised in a previous round of review and you feel that this manuscript is now acceptable for publication, you may indicate that here to bypass the “Comments to the Author” section, enter your conflict of interest statement in the “Confidential to Editor” section, and submit your "Accept" recommendation.

Reviewer #1: (No Response)

2. Is the manuscript technically sound, and do the data support the conclusions?

Reviewer #1: Yes

3. Has the statistical analysis been performed appropriately and rigorously? 

Reviewer #1: Yes

4. Have the authors made all data underlying the findings in their manuscript fully available?

Reviewer #1: Yes

5. Is the manuscript presented in an intelligible fashion and written in standard English?

Reviewer #1: Yes

6. Review Comments to the Author

Reviewer #1: In this Manuscript, Weise et al. mapped the thermal sensitivity of human nasal mucosa. They tested for thermal sensitivity different locations in the nose of 20 healthy subjects and found that thermal threshold was different depending from the site of where the measures were taken. They also found several correlations between thermal sensitivity and olfactory function, ratings of trigeminal stimuli (AmmoLa® stick), a questionnaire regarding trigeminal function, nasal patency, and CO2 threshold. In addition, they attempt to identify molecular correlates to their finding by investigating the expression of TRP channels by performing qPCR from nasal swabs taken from different locations of human nasal mucosa.

The manuscript is interesting and present a complete characterization of human olfactory and trigeminal abilities.

I do have some concerns though:

1. It is not clear to me what could be the physiological meaning of testing thermal sensitivity in the nose. The authors might want to better explain the rationale of the study.

2. Since the most relevant and innovative aspect of the paper is indeed the nasal thermal stimulation, the authors should better explain the stimulation protocol. Indeed, it is not clear to me how the stimulation worked. For example: Could the authors apply different temperatures (from figure 2 it seems that this is the case but the discussion section lines 438 to 440 “thermal stimulation at various temperatures and for extended stimulating intervals could provide a more comprehensive characterization of perception” confused me)? In addition, how are duration and intensity related? I could not understand it from the methods section and figure 2 do not help in clarifying this issue.

3. A somehow related topic is the consideration of the thermal stimulation as mechanical one. This is reported in the discussion (line 426-431). Unless better explained I would consider thermal stimulation as different from mechanical. TRP channels are rather poorly sensitive to mechanical stimulation unless it is referred to a change in cellular volume. Do the authors mean that during heating the electrode went through thermal expansion? Would then be the expansion the mechanical stimulus? It is rather confusing; I think the issue could be solved by a clearer explanation of the set-up and protocol in the methods.

4. I think the qPCR lack of positive controls, since the authors state that often they failed to find TRPA1 and TRPM8. How could they be sure that the qPCR run under optimal conditions and with the right primers?

5. The Discussion section is too long, shortening may help to make it clearer.

7. PLOS authors have the option to publish the peer review history of their article (what does this mean?). If published, this will include your full peer review and any attached files.

Reviewer #1: No

---

## [Author Response · Author response to Decision Letter 1]

26 Apr 2024

Author´s reply to the comments of the reviewer

The authors thank the referees for the constructive criticism to which we would like to respond as follows:

1. It is not clear to me what could be the physiological meaning of testing thermal sensitivity in the nose. The authors might want to better explain the rationale of the study.

Answer:

The authors thank the reviewer for the comments. The olfactory and trigeminal system are closely interlinked in terms of anatomy and function. Trigeminal activation can be induced by thermal, mechanical, or chemosensory stimulation. While numerous studies have concentrated on mechanical and chemosensory trigeminal stimulation, there exists a gap in information concerning thermal trigeminal stimulation. This study seeks to fill this void by shedding light on the functional and physiological aspects of thermal stimulation within the nasal cavity. 

In this study, the thermal stimulation in the human nasal cavity was conducted for the first time. The stimulus threshold was investigated at various points within the nose. Additionally, the perception of suprathreshold stimulation was evaluated. Furthermore, nasal swabs were taken to examine receptor expression for the perception of thermal stimuli at different points within the nasal cavity.

The knowledge gained is important from a scientific perspective due to the inaugural exploration of thermal stimulation in the nasal cavity. Furthermore, these findings might be applied in the future for therapeutic trigeminal stimulation within the nose, for trigeminal dysfunction or olfactory dysfunction due to the close interlink between both systems.

2. Since the most relevant and innovative aspect of the paper is indeed the nasal thermal stimulation, the authors should better explain the stimulation protocol. Indeed, it is not clear to me how the stimulation worked. For example: Could the authors apply different temperatures (from figure 2 it seems that this is the case but the discussion section lines 438 to 440 “thermal stimulation at various temperatures and for extended stimulating intervals could provide a more comprehensive characterization of perception” confused me)? In addition, how are duration and intensity related? I could not understand it from the methods section and figure 2 do not help in clarifying this issue.

Answer:

The authors thank the reviewer for these comments. In response to the comment, the Methods section has been revised to clearly elucidate the procedure for thermal stimulation, see line 163-207.

3. A somehow related topic is the consideration of the thermal stimulation as mechanical one. This is reported in the discussion (line 426-431). Unless better explained I would consider thermal stimulation as different from mechanical. TRP channels are rather poorly sensitive to mechanical stimulation unless it is referred to a change in cellular volume. Do the authors mean that during heating the electrode went through thermal expansion? Would then be the expansion the mechanical stimulus? It is rather confusing; I think the issue could be solved by a clearer explanation of the set-up and protocol in the methods.

Answer:

The authors thank the reviewer for these comments. The reviewers are absolutely correct in pointing out the distinction between thermal and mechanical stimulation in this study. In the present study, the thermal stimulation was administered via an electrode that needs to be placed on the tissue. By positioning the electrode on the intranasal mucosa, discrete irritation can occur, which might result in a discrete mechanical stimulus. However, the placement of the electrode was done under endoscopic control in order to decrease the potential mechanical stimulation. We did not consider a thermal expansion due to the short exposure to heat. The potential mechanical irritation mentioned in the discussion (line 430) refers solely to the process of electrode placement on the tissue. The section methodology was revised to elucidate this aspect (see line 186-188). 

This study is the first one on thermal intranasal stimulation, emphasizing the significance of the study.

4. I think the qPCR lack of positive controls, since the authors state that often they failed to find TRPA1 and TRPM8. How could they be sure that the qPCR run under optimal conditions and with the right primers?

Answer:

The authors thank the reviewer for these comments. This study exclusively utilized assays validated by the manufacturer, which were cultivated under the conditions specified by the manufacturer. A positive control was conducted with the cell line LNCaP.

5. The Discussion section is too long, shortening may help to make it clearer.

Answer:

The authors thank the reviewer for these comments. The authors followed the reviewer’s suggestion, the section discussion was shortened and thoroughly revised (line 389-536).

---

## [Decision Letter · Decision Letter 2]

21 May 2024

Hot topic: Mapping of the human intranasal mucosal thermal sensitivity: A clinical study on thermal threshold and trigeminal receptors

PONE-D-23-24390R2

Dear Dr. Weise,

We’re pleased to inform you that your manuscript has been judged scientifically suitable for publication and will be formally accepted for publication once it meets all outstanding technical requirements.

Kind regards,

Johannes Reisert

Academic Editor

PLOS ONE

Additional Editor Comments (optional):

Reviewers' comments:

Reviewer's Responses to Questions

**Comments to the Author**

1. If the authors have adequately addressed your comments raised in a previous round of review and you feel that this manuscript is now acceptable for publication, you may indicate that here to bypass the “Comments to the Author” section, enter your conflict of interest statement in the “Confidential to Editor” section, and submit your "Accept" recommendation.

Reviewer #1: All comments have been addressed

2. Is the manuscript technically sound, and do the data support the conclusions?

Reviewer #1: Yes

3. Has the statistical analysis been performed appropriately and rigorously? 

Reviewer #1: Yes

4. Have the authors made all data underlying the findings in their manuscript fully available?

Reviewer #1: Yes

5. Is the manuscript presented in an intelligible fashion and written in standard English?

Reviewer #1: Yes

6. Review Comments to the Author

Reviewer #1: The authors adequately addressed my comments. In particular, the method of thermal stimulation is now better described.

7. PLOS authors have the option to publish the peer review history of their article (what does this mean?). If published, this will include your full peer review and any attached files.

Reviewer #1: No

---

## [Editor Report · Acceptance letter]

7 Jun 2024

PONE-D-23-24390R2 

PLOS ONE

Dear Dr. Weise, 

I'm pleased to inform you that your manuscript has been deemed suitable for publication in PLOS ONE. Congratulations! Your manuscript is now being handed over to our production team.

Kind regards, 

on behalf of

Dr. Johannes Reisert 

Academic Editor

PLOS ONE